# Comprehensive, structurally-informed alignment and phylogeny of vertebrate biogenic amine receptors

Stephanie J. Spielman, Keerthana Kumar and Claus O. Wilke

Department of Integrative Biology, The University of Texas at Austin, Austin, USA
Institute of Cellular and Molecular Biology, The University of Texas at Austin, Austin, USA
Center for Computational Biology and Bioinformatics, The University of Texas at Austin, Austin, USA

Corresponding author
Stephanie J. Spielman,
stephanie.spielman@gmail.com

## ABSTRACT

Biogenic amine receptors play critical roles in regulating behavior and physiology in both vertebrates and invertebrates, particularly within the central nervous system. Members of the G-protein coupled receptor (GPCR) family, these receptors interact with endogenous bioamine ligands such as dopamine, serotonin, and epinephrine, and are targeted by a wide array of pharmaceuticals. Despite the clear clinical and biological importance of these receptors, their evolutionary history remains poorly characterized. In particular, the relationships among biogenic amine receptors and any specific evolutionary constraints acting within distinct receptor subtypes are largely unknown. To advance and facilitate studies in this receptor family, we have constructed a comprehensive, high-quality sequence alignment of vertebrate biogenic amine receptors. In particular, we have integrated a traditional multiple sequence approach with robust structural domain predictions to ensure that alignment columns accurately capture the highly-conserved GPCR structural domains, and we demonstrate how ignoring structural information produces spurious inferences of homology. Using this alignment, we have constructed a structurally-partitioned maximum-likelihood phylogeny from which we deduce novel biogenic amine receptor relationships and uncover previously unrecognized lineage-specific receptor clades. Moreover, we find that roughly 1% of the 3039 sequences in our final alignment are either misannotated or unclassified, and we propose updated classifications for these receptors. We release our comprehensive alignment and its corresponding phylogeny as a resource for future research into the evolution and diversification of biogenic amine receptors.

## INTRODUCTION

Biogenic amines such as the molecules serotonin and dopamine play critical roles in virtually all Metazoans and exert significant influence on both behavior and physiology. In vertebrates, the biogenic amine receptor family, which includes dopamine (DRD), histamine (HRH), trace (TAAR), adrenergic (ADR), muscarinic cholinergic (mAChR),

and most serotonin (5HTR) receptors, primarily mediates biogenic amine activity. Biogenic amine receptors belong to the broad family of G protein-coupled receptors (GPCRs), one of the largest and most diverse eukaryotic receptor families. Indeed, due to the extensive diversity of biological functions they direct and the ongoing expansion of their ligand repertoire, GPCRs are considered one of the most evolutionarily innovative and successful gene families (*Bockaert & Pin, 1999*; *Lagerstrom & Schioth, 2008*).

Biogenic amine receptors form a clade within the large Rhodopsin-like GPCR family (*Fredriksson et al., 2003*; *Kakarala & Jamil, 2014*), whose emergence likely accompanied that of the Opisthokont (Fungi and Metazoa) lineage (*Krishnan et al., 2012*). The Rhodopsin-like family expanded substantially in Metazoa, and the specific diversification of biogenic amine receptors has contributed significantly to the functioning of the central nervous system (*Callier et al., 2003*; *Nichols & Nichols, 2008*). Like all GPCRs, biogenic amine receptors have a characteristic, highly-conserved structure of seven transmembrane (TM) domains separated by three extracellular (ECL) and three intracellular (ICL) loops, and they propagate intracellular signaling through a G-protein-mediated pathway. Moreover, these receptors are prominent targets for a wide range of pharmaceuticals aimed to treat myriad diseases such as schizophrenia, migraines, hypertension, allergies and asthma, and stomach ulcers (*Schoneberg et al., 2004*; *Evers et al., 2005*; *Mason et al., 2012*).

In spite of the biological and clinical importance of these receptors, studies on their evolution are limited and have predominantly focused on individual receptor subtypes, namely TAAR (*Gloriam et al., 2005*; *Lindemann et al., 2005*; *Hashiguchi & Nishida, 2007*), DRD (*Callier et al., 2003*; *Yamamoto et al., 2013*), and 5HTR (*Anbazhagan et al., 2010*). Moreover, many of these studies, and indeed studies on the general evolution of the Rhodopsin-like family, have examined very narrow species distributions; specifically teleosts (*Gloriam et al., 2005*), primates (*Anbazhagan et al., 2010*), humans and mice (*Vassilatis et al., 2003*; *Kakarala & Jamil, 2014*), or even strictly humans (*Fredriksson et al., 2003*). Thus, virtually no studies have been conducted that account for the full breadth of vertebrate biogenic amine receptor sequences.

To gain a comprehensive understanding of this receptor family's evolution, a robust multiple sequence alignment (MSA) is needed. MSAs provide the foundation for nearly all comparative sequence analyses, and they are commonly used to locate conserved sequence motifs, identify functionally important residues, and investigate evolutionary histories. Since constructing an MSA represents the first step in any sequence analysis, MSA errors are known to bias these downstream analyses (*Ogden & Rosenberg, 2006*; *Wong, Suchard & Huelsenbeck, 2008*; *Jordan & Goldman, 2012*). It is therefore crucial to ensure accuracy in MSAs to the extent possible.

For GPCR sequences in particular, any MSA should recapitulate the canonical seven-TM structure, which a naive alignment of sequences cannot necessarily accomplish. Several varieties of MSA software platforms have been developed that incorporate structural information into the alignment algorithm by aligning sequences to a given protein crystal structure (*Pei, Kim & Grishin, 2008*; *O'Sullivan et al., 2004*) or hidden Markov model (HMM) profile (*Eddy, 1998*; *Chang et al., 2012*; *Hill & Deane, 2012*). In fact,

some programs, such as MP-T (*Hill & Deane, 2012*) and TM-Coffee (*Chang et al., 2012*), cater specifically to membrane proteins. However, all of these programs are extremely computationally-intensive and thus ill-suited for large-scale applications. Furthermore, many of these programs require the use of a single crystal structure or HMM profile to guide sequence alignment. While all GPCRs contain seven TM domains, different GPCR subfamilies, particularly the biogenic amine receptors, feature a wide variety of ICL and ECL sizes. For example, human HRH1 and DRD3 contain roughly 27 and 117 residues, respectively, in their ECL3 domains, and roughly 68 and 14 residues, respectively, in their ICL3 domains (as predicted by GPCRHMM (*Wistrand, Käll & Sonnhammer, 2006*)). Thus, aligning diverse sequences using a single structure may not effectively capture the domain variability across biogenic amine receptor subtypes. Instead, a desirable alignment strategy would anchor all sequences by their conserved TM domains without inappropriately constraining the heterogeneous ECL and ICL domains.

To this end, we integrated a traditional progressive alignment approach with robust structural predictions to generate a structurally-informed, comprehensive (3039 sequences) MSA of vertebrate biogenic amine receptors, representing the most extensive such dataset to date. Through this strategy, we overcame the computational limitations imposed by explicitly structurally-aware MSA construction platforms while still incorporating structural information into MSA construction. We used our structurally-informed MSA to build a maximum likelihood (ML) phylogeny of vertebrate biogenic amine receptors, and we found that a partitioned phylogeny which separately considered TM and extramembrane (EM) domains dramatically improved phylogenetic fit relative to an unpartitioned phylogeny. Using this structurally-partitioned phylogeny, we were able to discern relationships among biogenic amine receptor subtypes with a far increased level of sensitivity relative to previous studies, as well as identify novel lineage-specific receptor clades and clarify NCBI annotations for over 30 sequences.

We present this vertebrate biogenic amine receptor MSA and its corresponding phylogeny as a resource for any group interested in studying the dynamic evolutionary processes and structural and/or functional constraints operating within this class of GPCRs. All data, including MSAs, phylogenies, and sequence descriptions, as well as all code used to generate these data, are freely available from https://github.com/sjspielman/amine_receptors. We expect that these data will prove useful for studying both the broad patterns governing biogenic amine receptor sequence evolution and the evolutionary trends specific to certain receptor subtypes. Further, our MSA should serve as a helpful resource in the ongoing development of homology models and pharmaceutical therapeutics targeting these receptors (*Kristiansen, 2004*; *Ishiguro, 2004*; *Evers et al., 2005*; *Mason et al., 2012*).

## METHODS AND MATERIALS

### Sequence collection and processing

We collected protein sequences using PSI-BLAST (*Altschul et al., 1997*), specifically from the RefSeq (v2.2.29+) database (*Pruitt et al., 2013*), for 42 distinct human biogenic amine

protein sequences representing the full range of known such receptors in the human genome. To obtain distant yet well-supported orthologs, we ran each PSI-BLAST search for 5 iterations with an e-value cutoff of $10^{-20}$, a sequence identity threshold of 25%, and a length difference of $\pm 50\%$ relative to the seed sequence. After combining all sequences recovered from the individual PSI-BLAST searches, we discarded duplicate sequences, leaving a total of 4232 PSI-BLAST results. We then filtered this sequence set to remove sequences from non-vertebrate taxa, sequences annotated as low-quality, pseudogene, and/or partial, and sequences which contained more than 1% ambiguous residues (i.e., B, X, or Z). Additionally, we used the program GPCRHMM (*Wistrand, Käll & Sonnhammer, 2006*) to determine whether a given sequence was indeed a GPCR. We discarded sequences which had either a local or global GPCRHMM score of less than 10: both extremely conservative thresholds. Thus, while it is possible that some true GPCRs were discarded, these stringent thresholds for both local and global scores provide high confidence that all retained sequences were indeed GPCRs. Together, these filters left a total of 3464 receptor sequences.

## Sequence alignment and phylogenetic reconstruction

Before aligning the sequences, we used the program GPCRHMM (*Wistrand, Käll & Sonnhammer, 2006*) to assign each residue in all protein sequences to its respective structural domain (extracellular, transmembrane, or intracellular) using a 0.5 posterior probability cutoff. We then aligned and filtered sequences according to the strategy outlined in Fig. 3, which specifically employed MAFFT v7.149b using the default algorithm (*Katoh & Standley, 2013*).

All phylogenies were created using RAxML v8.1.1 (*Stamatakis, 2014*) using the LG (*Le & Gascuel, 2008*) amino acid exchangeability matrix with empirical amino acid frequencies (+F) and the CAT model of site heterogeneity (*Stamatakis, 2006*), with the default 25 rate categories. For inferences incorporating structural partitions, we assigned each partition a unique evolutionary model using these settings, thus allowing each partition to have a distinct equilibrium amino-acid frequency and rate distributions. Final parameter values for all phylogenetic inferences were optimized with the GAMMA model of heterogeneity. We performed 100 phylogenetic inferences for each parameterization to thoroughly search the tree space, and we used a likelihood ratio test (LRT) to compare the best resulting trees from each parameterization. We computed 200 bootstrapped trees using RAxML for each resulting phylogeny.

## RESULTS AND DISCUSSION

### Constructing a structurally-informed MSA of biogenic amine receptors

We collected all biogenic amine receptor sequences from the RefSeq database (*Pruitt et al., 2013*) using PSI-BLAST and removed all poor-quality sequences (see 'Methods' for details). We then used the software GPCRHMM (*Wistrand, Käll & Sonnhammer, 2006*) to identify whether each sequence in our data set was indeed a GPCR. GPCRHMM uses

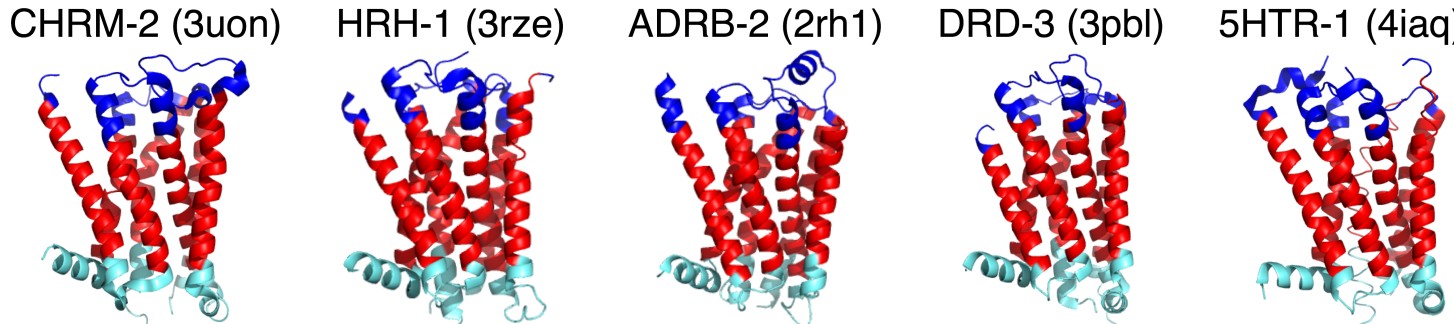

**Figure 1 GPCRHMM domain predictions for representative biogenic amine receptor crystal structures from the Protein Data Bank (PDB).** Gene names are shown in capital letters above each structure, and corresponding PDB IDs are shown in parentheses. Dark blue represents predicted extracellular residues, red represents predicted TM residues, light blue represents predicted intracellular residues.

a hidden Markov model approach to identify GPCRs from protein sequence data alone, and features an exceptionally low false positive rate (∼1%) as well as a 15% increase in sensitivity relative to other similar structural prediction programs (*Wistrand, Käll & Sonnhammer, 2006*). We removed all sequences which GPCRHMM could not robustly classify as a GPCR, leaving a dataset of 3464 protein sequences.

In addition to identifying GPCR sequences, GPCRHMM can also predict transmembrane domain regions for Rhodopsin-like GPCRs with exceptional accuracy (*Spielman & Wilke, 2013*). As shown in Fig. 1, GPCRHMM yields excellent domain predictions for resolved biogenic amine receptor crystal structures and thus serves as a robust proxy for more computationally-intensive structural predictors. Therefore, we used GPCRHMM to predict the structural domain (TM, extracellular, or intracellular) for each residue in these sequences.

To begin, we aligned all 3464 protein sequences with MAFFT (*Katoh & Standley, 2013*) with default settings. Using GPCRHMM's domain predictions, we determined the consensus structural domain for each MSA column to assess how well the MSA recapitulated the overarching GPCR domain structure (Fig. 2A). Although we had already filtered out putatively non-GPCR sequences, we found that several hundred sequences did not align according to the overarching domain structure. Many sequences were shifted out of structural frame, causing TM domains to inappropriately align with loop domains, or vice versa. Moreover, the mere presence of these misaligned sequences in the naive MSA introduced a substantial amount of gaps, oftentimes within a single TM domain (notably TM1 and TM7, as seen in Fig. 2A). While gaps are not inherently problematic in MSAs, the strong evolutionary pressures conserving the GPCR structure should prevent large indel events from occurring within TM domains. Thus, many of the large gaps in this alignment were likely produced by the presence of confounding sequences in the dataset.

We therefore adopted an iterative strategy integrating a progressive sequence alignment with the GPCRHMM structural predictions to systematically cull poorly-aligned sequences. As outlined in Fig. 3, we first aligned protein sequences with MAFFT (*Katoh & Standley, 2013*). Using the residue domain assignments computed with GPCRHMM, we determined the consensus domain for each column in this MSA. Next, we discarded
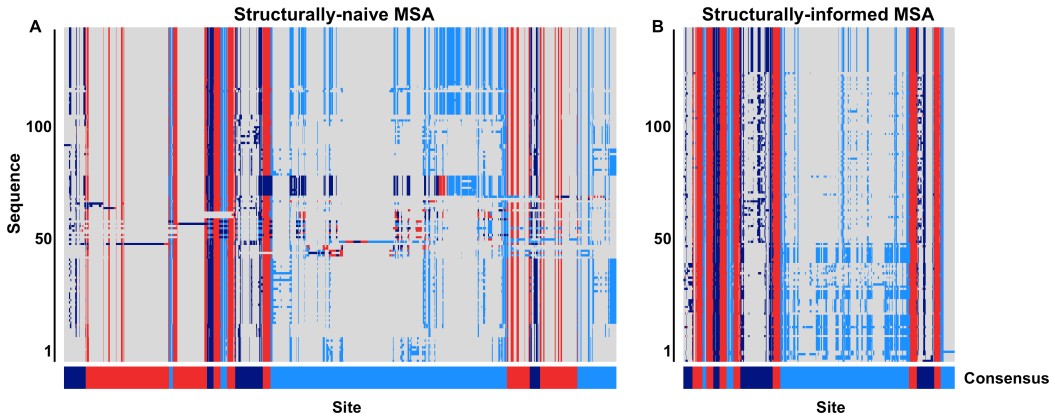

**Figure 2 Graphical representation of a subset of the (A) structurally-naive and (B) structurally-informed biogenic amine receptor MSAs.** Each panel displays 130 MSA rows focused specifically on the MSA section containing the seven TM domains. Dark blue represents predicted extracellular residues, red represents predicted TM residues, lighter blue represents predicted intracellular residues, and gray represents MSA gaps. The bottom bar below each MSA figure shows the consensus domain structure for each MSA. The structurally-naive MSA was built all 3464 putative GPCR sequences in MAFFT, whereas the structurally-informed MSA was built using the iterative strategy outlined in Fig. 3. Note that all columns which contain only gaps in this MSA subset have been removed from this figure for visual clarity.

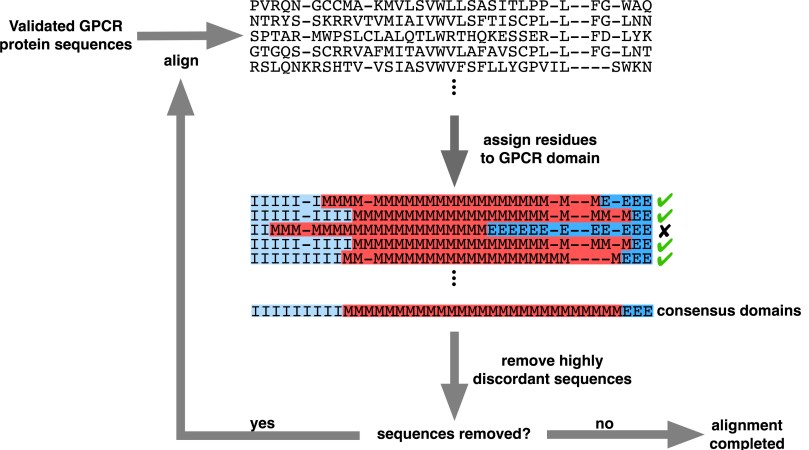

**Figure 3 Iterative alignment strategy used to generate the structurally-informed vertebrate biogenic amine receptor MSA.** A total of 3464 sequences were initially input ("Validated GPCR protein sequences"), and the final structurally-informed MSA contained 3039 protein sequences. Residues marked with "I" represent intracellular residues, those marked with "M" represent TM residues, and those marked with "E" represent extracellular residues. MSA gaps were treated as missing data when determining each column's consensus structural domain. Sequenced were removed ("remove highly discordant sequences") if $\geq$5% of columns belonged to a different structural domain than the respective consensus domain. Note that the MSA shown in this figure represents a subset of the entire MSA.

all sequences for which $\geq$5% of residues did not correspond to their respective column's consensus domain. We realigned the remaining sequences with MAFFT and continued in this manner until no more sequences were discarded. Importantly, this strategy did not require any manual data filtering or visual data inspection, thus avoiding any

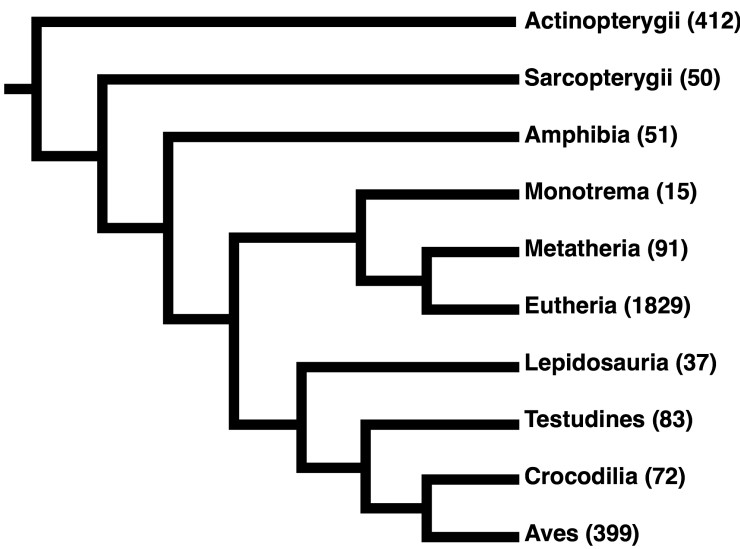

**Figure 4 Cladogram of the taxonomic distribution of all sequences in the final structurally-informed MSA.** All sequences belonged to the Euteleostomi clade of jawed vertebrates. Numbers in parentheses indicate the total number of sequences from the respective clade. We note that our MSA was particularly enriched for sequences from Eutherian (placental mammal) species, likely due to the stringent filters we applied to sequence collection that favored fully-sequenced genomes.

**Table 1 Biogenic amine receptor classes, and their abbreviations, considered in this study.** The receptor class "Unknown" refers to the corresponding uncharacterized clade in Fig. 5, and the column "N" indicates the total number of sequences for each broad receptor class in our structurally-informed MSA.

| Receptor class | Abbreviation | N |
| --- | --- | --- |
| Serotonin receptors | 5HTR | 972 |
| Adrenergic receptors | ADR | 611 |
| Dopamine receptors | DRD | 464 |
| Muscarinic cholinergic receptors | mAChR | 353 |
| Trace amine-associated receptors | TAAR | 343 |
| Histamine receptors | HRH | 286 |
| Unknown receptors | Unknown | 10 |

confounding subjectivity in MSA processing. Ultimately, this strategy produced a final MSA which indeed captured the conserved GPCR domain structure, as depicted in Fig. 2B. This structurally-informed MSA contained 3039 sequences broadly distributed across receptor subtypes (Table 1) and vertebrate taxa (Fig. 4). Additionally, we noted that the structurally-informed MSA contained canonical Rhodopsin-family GPCR motifs, including the E/DRY motif at the boundary of TM3 and ICL2 (MSA columns 2351-3) and the NPxxY motif at the boundary of TM7 and the C-terminal loop (MSA columns 3411-5).

## Structurally-aware MSA strongly improves phylogenetic fit

Next, we used this structurally-informed MSA to infer a maximum likelihood (ML) phylogeny of vertebrate biogenic amine receptors in RAxML. Previous work has shown

that combined structural and functional constraints impose differing selection pressures in TM vs. EM domains, in turn producing distinct amino-acid frequencies and evolutionary rates in each domain class (*Tourasse & Li, 2000*; *Stevens & Arkin, 2001*; *Julenius & Pedersen, 2006*; *Oberai et al., 2009*; *Spielman & Wilke, 2013*; *Franzosa, Xue & Xia, 2013*). As our MSA allowed us to confidently identify each MSA column as either TM or EM, we were able to conduct a more rigorous phylogenetic inference using a partitioned analysis. Therefore, we inferred two ML phylogenies: one with two partitions representing TM and EM columns, respectively, and one with a single partition for the entire MSA. The former scheme allowed each partition to have unique distributions of evolutionary rate heterogeneity and stationary amino-acid frequencies, thus accounting for the distinct selective regimes in each domain. To ensure as much as possible that the EM and TM partitions contained only residues belonging to their respective structural domain, we created a masked MSA in which protein residues which did not conform to their respective consensus domains were replaced with a question mark. All phylogenetic analyses were conducted using this masked MSA.

We performed 100 ML tree inferences for the unpartitioned and partitioned case each, and we compared the best resulting phylogeny from each scheme using the likelihood ratio test. We found that the partitioned model offered dramatic improvements in phylogenetic fit ($p < 1^{-100}$), highlighting the benefits of analyzing GPCRs in a structurally-aware context. We thus proceeded to analyze the partitioned phylogeny more in depth.

## Structurally-aware phylogeny displays unknown biogenic amine receptor relationships and clades

Our resulting phylogeny, shown in Fig. 5, represents the most comprehensive vertebrate biogenic amine receptor phylogeny to date. This tree broadly captures many known features of biogenic amine receptor evolution, in particular that these receptors do not cluster based on ligand-binding but rather have undergone extensive functional convergent evolution. Indeed, our phylogeny indicates that only two ligand-based receptor classes, mAChR and TAAR, are truly monophyletic.

Our phylogeny features remarkably high bootstrap support for each distinct clade of receptor subtypes. We additionally find very strong support for three deep nodes that separate distinct receptor subtypes. The first contains the three clades HRH1, mAChR, and HRH-3,4, the second contains the clades 5HTR-1, 5HTR-5, and 5HTR-7, and the third contains the 5HTR-4 and TAAR clades. Previous studies have yielded conflicting phylogenetic placements for the 5HTR-7 clade; some have argued that 5HTR-7 is phylogenetically distinct from all other 5HTR sequences (*Kakarala & Jamil, 2014*), while others have found evidence for a single clade containing 5HTR-5,7 as a sister taxa to a clade containing ADRA1 sequences (*Fredriksson et al., 2003*). Alternatively, we find moderate-to-strong support for the 5HTR-7 clade having originated before subsequent diversification into 5HTR-5 and 5HTR-1, and we find full support in favor of ADRA1 forming an entirely distinct monophyletic group outside all other vertebrate biogenic amine receptors. Additionally, HRH-3,4 appears as a single monophyletic group.

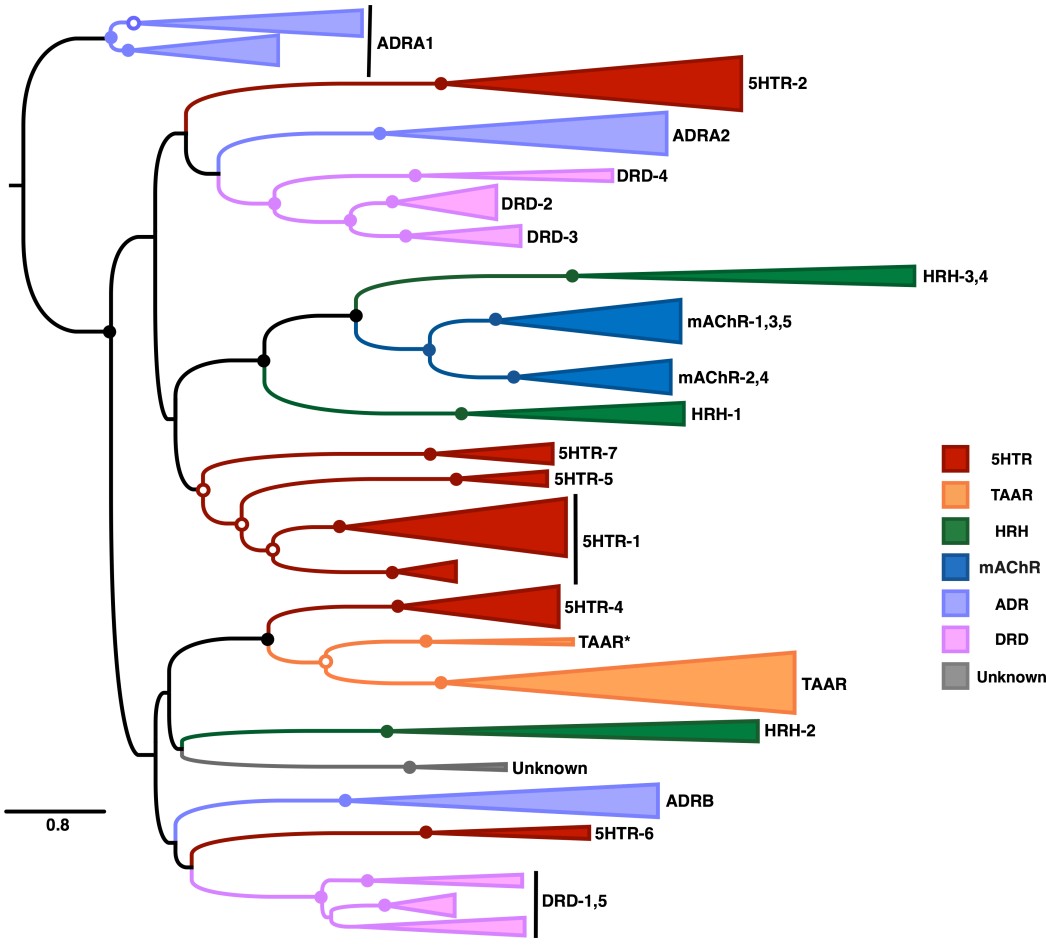

**Figure 5 Maximum-likelihood phylogeny of vertebrate biogenic amine receptors built using the masked structurally-informed MSA in RAxML.** Nodes with open circles indicate ≥50% bootstrap support, and nodes with closed circles indicate ≥90% bootstrap support. Biogenic amine receptors are abbreviated as in Table 1. The clade labeled "Unknown" could not be clearly identified as one of the major receptor types and may represent a previously unrecognized biogenic amine receptor clade. Note that the root shown on this tree has been placed arbitrarily.

While the HRH-4 clade contains only mammalian sequences, including monotreme (platypus) sequences, HRH-3 sequences are broadly distributed across vertebrate taxa. This taxonomic distribution suggests that HRH-4 is a mammalian-specific histamine receptor class that arose from an HRH-3 duplication concurrent with mammalian origins.

In addition, among the 3039 sequences in the structurally-informed MSA we identified 31 sequences (∼1% of our dataset) that we considered misannotated (Table 2), either because the NCBI annotation did not match the phylogenetic placement of the sequences or because the sequences did not cluster with known biogenic amine receptor types. Several NCBI annotations identified the correct receptor class but the incorrect receptor subtype, whereas other sequences were entirely uncharacterized. In particular, we identified an entirely unknown clade of biogenic amine receptors. This unknown clade, which appears as a sister to HRH2 in Fig. 5, only contains avian sequences and a single *Xenopus tropicalis*

**Table 2 Misannotated and uncharacterized sequences identified through phylogenetic analysis.** Based on sequence placement in the structurally-partitioned phylogeny, we propose updated classifications for 31 biogenic amine receptor sequences. The proposed classifications "Unknown" and "TAAR*" refer to the correspondingly-labeled clades in Fig. 5.

| Protein ID | Nucleotide ID | Current classification | Proposed classification |
| --- | --- | --- | --- |
| XP_005797918.1 | XM_005797861.1 | DRD-2 | DRD-3 |
| XP_003967971.1 | XM_003967922.1 | DRD-2 | DRD-3 |
| NP_001266433.1 | NM_001279504.1 | mAChR-4 | mAChR-2 |
| XP_001520508.2 | XM_001520458.3 | HRH-3 | HRH-4 |
| XP_005282846.1 | XM_005282789.1 | HRH-4 | HRH-3 |
| XP_001920844.1 | XM_001920809.1 | TAAR-4-like | TAAR-12 |
| NP_001076571.1 | NM_001083102.1 | TAAR-64 | TAAR-13 |
| XP_006014096.1 | XM_006014034.1 | TAAR-9-like | TAAR-4 |
| XP_003201718.2 | XM_003201670.2 | TAAR-1-like | TAAR-10 |
| NP_001076546.1 | NM_001083077.1 | TAAR-11-like | TAAR-10 |
| NP_001083418.1 | NM_001089949.1 | uncharacterized | ADRB |
| NP_001103208.1 | NM_001109738.1 | uncharacterized | HRH-2 |
| NP_001124143.1 | NM_001130671.1 | uncharacterized | TAAR-12 |
| XP_001337671.1 | XM_001337635.2 | 5HTR-4-like | TAAR* |
| XP_003976403.1 | XM_003976354.1 | 5HTR-4-like | TAAR* |
| XP_005810466.1 | XM_005810409.1 | 5HTR-4-like | TAAR* |
| XP_003454279.1 | XM_003454231.1 | 5HTR-4-like | TAAR* |
| XP_004549625.1 | XM_004549568.1 | 5HTR-4-like | TAAR* |
| XP_002935532.2 | XM_002935486.2 | 5HTR-4-like | TAAR* |
| XP_006013317.1 | XM_006013255.1 | 5HTR-4-like | TAAR* |
| XP_005510029.1 | XM_005509972.1 | 5HTR-7-like | Unknown |
| XP_002187301.2 | XM_002187265.2 | Octopamine receptor-like | Unknown |
| XP_002937327.2 | XM_002937281.2 | Octopamine receptor-like | Unknown |
| XP_005045681.1 | XM_005045624.1 | Octopamine receptor-like | Unknown |
| XP_005144673.1 | XM_005144616.1 | Octopamine receptor-like | Unknown |
| XP_005229932.1 | XM_005229875.1 | Octopamine receptor-like | Unknown |
| XP_005428400.1 | XM_005428343.1 | Probable GPCR No9-like | Unknown |
| XP_005490920.1 | XM_005490863.1 | Probable GPCR No9-like | Unknown |
| XP_005518128.1 | XM_005518071.1 | Probable GPCR No9-like | Unknown |
| XP_006111669.1 | XM_006111607.1 | Octopamine receptor-like | Unknown |
| XP_420867.2 | XM_420867.4 | Octopamine receptor | Unknown |

(western-clawed frog) sequence. Thus, it is likely that this clade emerged concurrently with tetrapods and was secondarily lost in reptiles/birds and mammals. Interestingly, all but one of this clade's sequences were annotated in NCBI as either octopamine or No9-like receptors, both of which are insect-specific biogenic amine receptors that do not occur in vertebrate taxa (*Roeder, 2005*). The last sequence, alternatively, was annotated as 5HTR-7-like. Taken together, these sequence misannotations suggest an intriguing hypothesis that this clade evolved from an ancestral 5HTR sequence, and subsequent convergent evolution to insect-specific biogenic amine receptors has allowed these receptors to interact with atypical ligands for vertebrates.

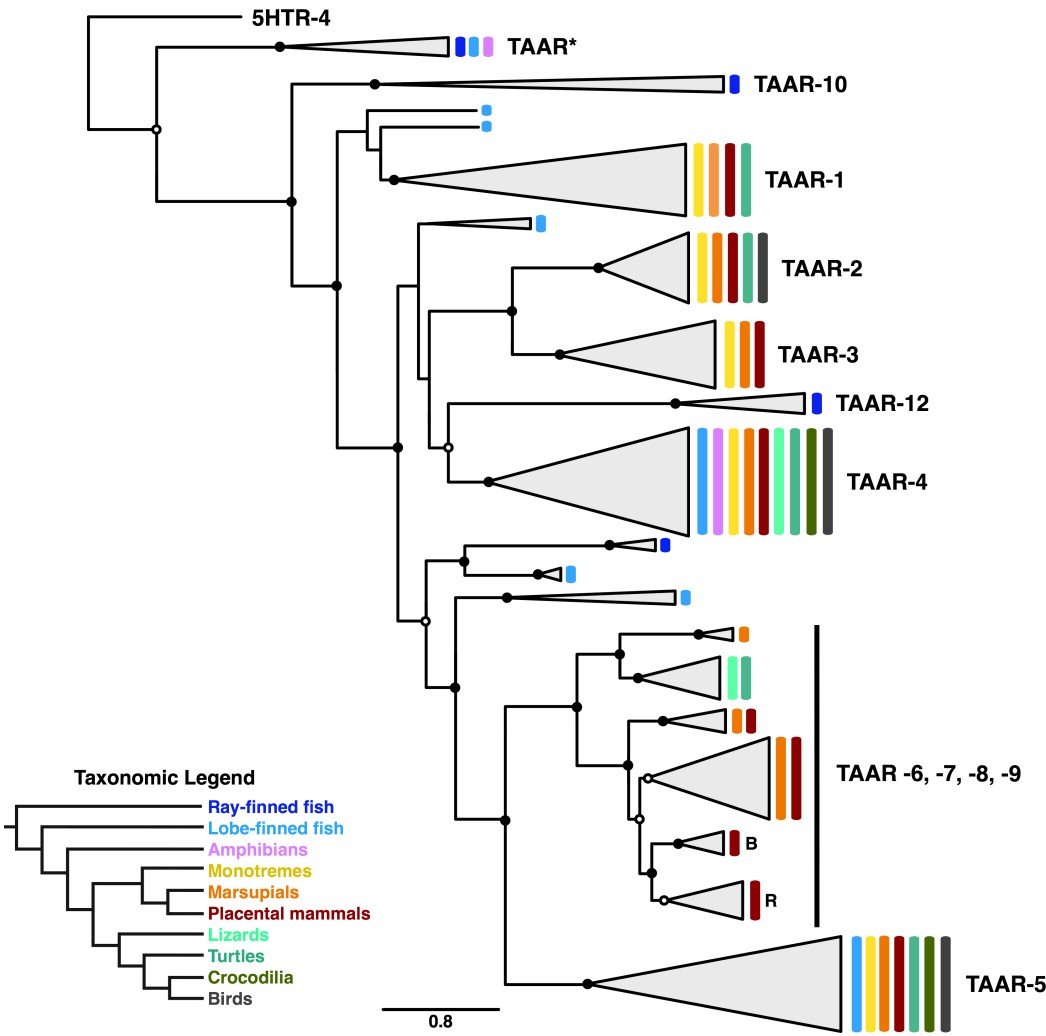

**Figure 6 Subclade of the TAAR receptors within the phylogeny shown in Fig. 5.** Nodes with open circles indicate ≥50% bootstrap support, and nodes with closed circles indicate ≥90% bootstrap support. The subclades within the TAAR-6,7,8,9 clade labeled as "B" and "R" indicate clades containing only bovid and rodent sequences, respectively.

## Dynamic lineage-specific evolution of the trace-amine associated receptors

Of particular interest in our phylogeny are the unique evolutionary patterns in the TAAR clade. While all TAAR sequences do cluster together, the TAAR phylogeny is consistent with previous suggestions that extensive expansion and contraction events have characterized this receptor family's evolution (*Lindemann et al., 2005*; *Hashiguchi & Nishida, 2007*; *Stäubert et al., 2010*; *Stäubert, Le Duc & Schöneberg, 2013*). In fact, the TAAR subtree, displayed in Fig. 6, differs somewhat from previously proposed TAAR phylogenies (*Lindemann et al., 2005*; *Hashiguchi & Nishida, 2007*) and indicates the presence of many lineage-specific subclades.

Interestingly, only a single clade in the TAAR phylogeny, comprised of the two subclades TAAR-4 and TAAR-12, contains representatives from the full species distribution considered in our study. Further, the close phylogenetic relationship between these two subclades suggests that TAAR-4 and TAAR-12 are in fact orthologous TAAR subtypes, and the distinct naming for these subtypes apparently emerged simply because TAAR-12 is ray-finned specific. In contrast to this group, all other clades contain distinct, limited species distributions, indicative of repeated gain and loss events. In particular, the broad clade containing subtypes TAAR-6, −7, −8, and −9 appears to have undergone substantial lineage-specific evolution, with certain subclades only present in marsupial and placental mammals and others only present in lizards and turtles. Moreover, we identified two small clades within the broad TAAR-6, −7, −8, −9 clade that apparent indicate lineage-specific expansions specifically within bovids (labeled "B" in Fig. 6) and rodents (labeled "R" in Fig. 6), respectively.

Throughout the TAAR phylogeny, ray-finned and lobe-finned fish sequences frequently appear as outgroups to tetrapod-specific clades, indicating progressive diversification tracking large-scale speciation events. However, we do note that several lobe-finned fish (coelacanth) sequences are scattered across the TAAR tree and do not clearly cluster with any TAAR subtypes, likely reflecting this lineage's ancient divergence and unique evolutionary trajectory (*Amemiya et al., 2013*). Moreover, amphibian sequences are notably absent from this phylogeny, relative to other taxonomic groups. While absence of such sequences in our data set does not necessarily imply that these genes have actually been lost in amphibians, such a hypothesis would be consistent with the overarching gain and loss patterns that TAAR sequences display and thus may merit further study.

We additionally identified a small clade sister to TAAR (labeled in Figs. 5 and 6 as TAAR*) that only contains sequences annotated by NCBI as "5HTR-4-like." At first glance, these annotations might suggest that 5HTR-4 is in fact paraphyletic, diversifying gradually before giving rise to TAARs. However, as all sequences in TAAR* belong taxonomically either to teleost or *Xenopus tropicalis*, we suspect that this clade actually corresponds to the so-called TAAR-V cluster identified by *Hashiguchi & Nishida (2007)*. Indeed, the TAAR-V cluster contains a similar taxonomic distribution to our TAAR* and constitutes an outgroup to all other vertebrate TAAR sequences, as our phylogeny similarly displays.

## Phylogenetic methods alone do not suffice to infer the evolutionary history of biogenic amine receptors

Although our phylogeny suggests several new features of biogenic amine receptor evolution, the majority of deeper splits in the phylogeny had very low bootstrap support, meaning that most of the broader relationships among biogenic amine receptors remain unresolved. This result highlights that a strictly phylogenetic approach cannot fully elucidate the complex evolutionary histories of expanding gene families. In particular, modern phylogenetic methods focus solely on the substitution process and treat MSA gaps simply as missing data. However, gaps actually represent insertion and deletion evolutionary events, and some have suggested that ignoring this information ultimately

hinders phylogenetic accuracy (*Morrison, 2008*; *Loytynoja & Goldman, 2008*; *Warnow, 2012*; *Luan et al., 2013*).

This limitation is especially problematic for GPCRs. Following duplication events, GPCRs experience major indel events in their ICL and/or ECL domains, leading to dramatic shifts in loop domain sizes during the sub/neofunctionalization process. Unfortunately, the evolutionary intermediates that existed during these domain transitions have long-since disappeared from genomes, and there is no obvious way to infer the sequences of these missing links. Although the substitution process is key for understanding GPCR evolution, fully classifying relationships among GPCR families requires some understanding of how these radical domain changes occur. Therefore, additional approaches, such as syntenic analyses (*Sundstrom, Dreborg & Larhammar, 2010*; *Widmark et al., 2011*; *Yegorov & Good, 2012*; *Hwang et al., 2013*), combined with the phylogeny presented here should prove useful towards resolving the complete evolutionary history of vertebrate biogenic amine receptors.

## CONCLUSIONS

We have established a comprehensive, structurally-informed MSA of vertebrate biogenic amine receptors. We hope that this MSA, along with its ML phylogeny, will serve as a robust resource for future studies investigating these receptors evolutionary dynamics, structural/functional constraints operating within distinct receptor clades, or overarching patterns that govern biogenic amine receptor evolution. Future work may seek to combine the analyses we have performed here with syntenic or molecular clock analyses to elucidate origin of the receptors and precise evolutionary trajectories. Moreover, our MSA should prove useful in increasing accuracy in homology modeling and/or pharmaceutical development for these clinically important receptors (*Kristiansen, 2004*; *Ishiguro, 2004*; *Evers et al., 2005*; *Mason et al., 2012*).

## ACKNOWLEDGEMENTS

All computational resources were provided by the University of Texas at Austin's Center for Computational Biology and Bioinformatics (CCBB). We would like to thank Ahmad R. Sedaghat, MD, PhD for suggesting biogenic amine receptor evolution as a worthwhile study system.

### Funding

This work was supported in part by NIH grant R01 GM088344, ARO grant W911NF-12-1-0390, DTRA grant HDTRA1-12-C-0007, and NSF Cooperative Agreement No. DBI-0939454 (BEACON Center). The funders had no role in study design, data collection and analysis, decision to publish, or preparation of the manuscript.

## Grant Disclosures

The following grant information was disclosed by the authors:

NIH grant: R01 GM088344.

ARO grant: W911NF-12-1-0390.

DTRA grant: HDTRA1-12- C-0007.

NSF Cooperative Agreement: DBI-0939454.

## Competing Interests

Claus O. Wilke is an Academic Editor for PeerJ.

## Author Contributions

- Stephanie J. Spielman conceived and designed the experiments, performed the experiments, analyzed the data, contributed reagents/materials/analysis tools, wrote the paper, prepared figures and/or tables, reviewed drafts of the paper.
- Keerthana Kumar contributed reagents/materials/analysis tools.
- Claus O. Wilke wrote the paper, prepared figures and/or tables, reviewed drafts of the paper.

## Data Deposition

The following information was supplied regarding the deposition of related data:

GitHub: https://github.com/sjspielman/amine_receptors

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
