# Peer review of "Comprehensive, structurally-informed alignment and phylogeny of vertebrate biogenic amine receptors"

_PeerJ, doi:10.7717/peerj.773_

## Round 0.1 · original submission · Major Revisions

Your submission had now been reviewed by two highly qualified referees and myself. Overall, they are supportive of the work but describe a number of specific points that need to be addressed. I share this view. Please address the referees' comments in earnest.

A few additional points:

1) I think it's important to highlight the fact that "structure" is only indirectly informing your procedure—through predictions by GPCRHMM. Calling this "structurally-curated alignment" may convey the wrong idea. How about "HMM-informed alignment" so something along these lines?

2) By design, the retained sequences have consistently predicted segments according to GPCRHMM. It is therefore natural that the resulting alignments look more compact. However, it's unclear that compact is necessarily better—at least not in an evolutionary sense (aligned residues => residues evolved from a common ancestral residue). Insertions that are specific to particular sequences necessarily lead to additional columns that look mostly "empty", which may offend our sense of aesthetics, but they might well be correct and in any case have little impact on downstream analyses (such as tree inference).

3) I agree with reviewer 2 that the likelihoods obtained from different MSAs (and/or with or without masked residues) are not directly comparable. I also agree with him that bootstrap support would be more meaningful (though note that bootstrap support is a measure of random error but is blind to systematic error).

4) "By masking these positions with an ambiguous character, we ensured that each MSA column strictly contained residues belonging to the same structural domain." This is too strong. These are merely predicted to be from the same model; the words "strictly" and "ensure" implies certainty.

Reviewer 1 ·

Basic reporting

No comments

Experimental design

No comments

Validity of the findings

No comments

Comments for the author

In this manuscript, the authors describe the construction of a multiple sequence alignment and its corresponding phylogeny of vertebrate biogenic amine receptors. Their method relies on the prediction of the canonical seven-transmembrane structure of G protein-coupled receptors (GPCRs), which biogenic amine receptors belong to. In the results, the authors show that incorporating the structural predictions improves alignment accuracy over an alignment that does not consider these structural predictions. The authors also make the alignment and phylogeny available for download.

Overall the manuscript is well written with clearly-presented figures and very few typos.

Comments:

- The title is somewhat misleading in that the alignment is based on structure predictions rather that the experimentally determined structure of the receptors. In addition it is not curated but filtered as, aside from the initial collection of the sequences, there is no manual intervention in the process.

- The results rely utterly on the accuracy of Wistrand et al's GPRCHMM program. How accurate is this program? If this program has even a modest error rate, then so too will the alignments.

- Table 1 does not seem to be referenced in the text, aside from in the caption to Figure 4.

- Results and Discussion, 1st paragraph: "We collected all sequences..." Perhaps "We collected all biogenic amines sequences..." to ensure the reader does not think that all GPRC sequences were collected? Also, typo confidentally should be confidently?

- Results and Discussion, 2nd paragraph: does the determination of the consensus domains include the sequences about to be excluded? It appears the excluded sequences are included, as in Figure 1 the left-most 'M' in the consensus could be 2 or 3 out of 5 while the left-most 'E' could be 3 or 2 out of 5.

·

Basic reporting

The URL in the link at the end of the Introduction is wrong. It should be:
https://github.com/sjspielman/amine_receptors

Experimental design

The new method is based on alignment using MAFFT, and structural curation of the consistency of aligned structural domains: transmembrane, intra- and extra-cellular.

When proposing a new approach for sequence alignment, it is necessary to compare its performance to previously published methods. In the case of the present study, the new method can (and should) be compared to methods that were designed for structural-aware alignment, especially the consideration of TM domains, and even specifically for GPCR proteins. The study by Hopf et al. 2012 has done exactly that (doi: 10.1016/j.cell.2012.04.012), based on alignment by HHblits (Remmert et al. 2012, doi:10.1038/nmeth.1818). MP-T (doi:10.1093/bioinformatics/bts640) was also developed for structural-aware alignment of TM proteins, and was found to be more accurate than both MAFFT and HHblits. Another program is TM-Coffee (doi:10.1186/1471-2105-13-S4-S1). Therefore, the publication of a new method requires comparison of performance relative to these previous ones.

Which variant of MAFFT was used (e.g. L-INS-i?) and why?

In addition to masking residues that disagree with the structural domain prediction, badly aligned residues can be identified using tools such as HoT (http://www.ncbi.nlm.nih.gov/pubmed/18229673) and Guidance (doi: 10.1093/nar/gkq443), and if the aligner is switched - also T-Coffee (or TM-Coffee) consistency scores. It would be interesting to compare and combine the different predictors of alignment errors.

Page 3 paragraph 3: The conclusion that “our structurally-curated MSA featured far less error than did a structurally-naive MSA” is not supported by the presented results. Alignments with less gaps are not necessarily more accurate. Some alignments may be “over-aligned”, i.e. more compact than the true alignment. Also the prediction of secondary structure is not a strong indication of alignment quality, because it is merely a prediction. I suggest using actuall protein structures as a benchmark, which is the common practice in the field (e.g. Balibase and also in the abovementioned MP-T paper). There are multiple amine receptors in the PDB, which can be used for structural alignment. Then the accuracy of the proposed alignment can be assessed by comparison to these structural alignments.

“Structurally-aware MSA strongly improves phylogenetic inference”: The likelihood scores of trees built from the naive and structurally-curated alignments cannot be compared using AIC or in any other way because some of the sequences were removed in the filtering. Only likelihood scores of trees built for the same sequence data can be compared. Therefore, I suggest removing the same sequences also from the naive alignment (without re-aligning) and running raxml on that. The resulting tree of the same number of sequences will probably still be different from the tree that was built from the structurally-curated alignment. Then the likelihood scores may be compared.

Another way to show the superiority of the tree is if bootstrap scores are higher than the naive tree.

MrBayes is usually superior to ML algorithms such as RAxML. I suggest applying it to the best alignment. If the run time is too long because of the large number of sequences, then I suggest sampling a smaller number using CD-hit (http://bioinformatics.org/cd-hit/).

Page 5 last paragraph: what was different between the models in RAxML for the different partitions? For the TM domains, a TM-specific matrix should be used.

Validity of the findings

“or this clade represents an avian-specific diversification which the Xenopus tropicalis sequence resembles only convergently“ - convergence is very unlikely to mislead phylogeny reconstruction into joining neighboring clades that were not truly sister clades in their evolution. Also, if HRH-2 contains non-avian sequences than it is not possible that the new clade is an avian-specific diversification.

As in the specific discussion of the TAAR clade, it would be interesting to note which species are present in each of the other subfamilies, and thus infer an approximate dating of their origin. E.g. if some clades are tetrapod-specific then they may be the result of functional specialization that occurred after the divergences of tetrapods. Others that include also fish species are more ancient.

---

## Round 0.2 · Minor Revisions

Thank you for your revisions, and for addressing the referees's and my reservations. One reviewer is now fully satisfied, and the other has two outstanding concerns regarding the use of MAFFT with default parameters and the lack of Bayesian analysis. Both points are valid, but I am also cognisant of the amount of follow-up analyses this may represent, particularly the second point.

Therefore, I see two possible courses of action:

a) you could perform the additional controls requested by the reviewer, which would indubitably provide the support required for your assertions on the high quality of your alignment and phylogeny.

b) alternatively, you could weaken your assertions pertaining to tree and alignment quality. In particular, specifically drop claims of "high-quality" sequence alignment (several occurrences) and change your statement "our phylogeny reveals that only two ligand-based receptor classes, mAChR and TAAR, are truly monophyletic." into "our phylogeny indicates that ...".

Reviewer 1 ·

Basic reporting

No comments

Experimental design

No comments

Validity of the findings

No comments

Comments for the author

I am happy to say that the authors have fully addressed all the points I raised in my initial review

·

Basic reporting

No comments

Experimental design

In their response to the first review, the authors argue that the "the primary goal of our analysis is to generate a large, comprehensive dataset of biogenic amine receptors". Thus, they turn down several suggestions for improvement as “outside the scope of this study”. But this contradicts the title of the paper. Even after the change of wording it still says that the “structurally-informed alignment and phylogeny” are the main results of the analysis. Not just the collection of of a set of sequences (which is a trivial analysis). Therefore the choices of the alignment and the tree reconstruction algorithms are vital. The authors claim that it's not possible to align thousands of sequences by MAFFT L-INS-i. I'm not sure this is true. They should try to do so using a powerful computer. Note that since version 7.184 MAFFT supports multithreading, so using a powerful multi-core computer the running time can be reduced by a significant factor. Even if L-INS-i is indeed not applicable, there is a range of variants between L-INS-i and FFT-NS-1 (FFT-NS-2, FFT-NS-i). The most accurate alignment method which is still feasible should be used. One should expect a significant improvement even between FFT-NS-1 and FFT-NS-2 for such a large set of sequences (based on my own unpublished results and also Liu et al. Science 2009, DOI: 10.1126/science.1171243). And the difference in running time should be only about two fold.

Similarly, the choice of phylogeny reconstruction algorithm is critical for presenting the phylogenetic results of the paper. The authors should either remove the phylogenetic conclusions or attempt to apply the best phylogenetic method possible. It is true that a fully Bayesian analysis (e.g. MrBayes) cannot be applied to the full dataset. But the phylogenetic conclusions of the paper do not require the full 3000 sequences. As I wrote in my first review, a representative sample of all the major subtrees would be sufficient, and switching to a more accurate tree reconstruction method (e.g. MrBayes) would be much more important for the accuracy of the relationship between subtrees (the deeper branches). As for MAFFT, a parallelized version of MrBayes is available (both multi-threading and MPI are implemented) which allows utilizing large computer clusters to speed up the analysis by orders of magnitude, depending on the number of available servers in the cluster. Such clusters are present in even in small universities, and the University of Texas has some very large clusters.

“As shown in Figure 1, GPCRHMM yields excellent domain predictions for resolved biogenic amine receptor crystal structures and thus serves as a robust proxy for more computationally-intensive structural predictors.” - It is insufficient to present a few example for demonstrating the accuracy of prediction. One should present statistics for a representative benchmark dataset. This was already done by the authors of GPCRHMM, using a cross-validation approach. So I recommend to simply cite them. Furthermore, since GPCRHMM was trained only on proteins having a crystal structure in the database then it is expected to yield more accurate results for these proteins that served as training data and less accurate results for other proteins. This is because the HMM might have been over-fitted to some specific GPCR subtypes that were over-represented in the structural database. Now that you apply it to sequences from other subtypes you might experience lower accuracy than reported for the cross-validation analysis. This pitfall should be noted and discussed.

Validity of the findings

No comments

---

## Round 0.3 · accepted · Accept

Thank you for your prompt resubmission. I'm of course fine with you choosing the option of weakening your assertions pertaining to the alignment and tree, but this needs to be done more rigorously. For example, there remain 5 instances of the contentious "reveal" word usage, including one in a section title. Here are a couple of other strong assertions that require hedging:

- "we find full support showing that ADRA1 forms.." → we find full support in favour of ADRA1 forming..
- "We additionally found that HRH-3,4 is actually a single monophyletic group" → "In our phylogeny, HRH-3,4 appears as a single monophyletic group"
- "Although we were able to identify several new features of" → "Although our phylogeny suggests several new features"

To speed things up, I am accepting the manuscript, but please fix these remaining issues during the proofing stage.